# Preventive Effects of the Marine Microalga *Phaeodactylum tricornutum*, Used as a Food Supplement, on Risk Factors Associated with Metabolic Syndrome in Wistar Rats

**DOI:** 10.3390/nu11051069

**Published:** 2019-05-14

**Authors:** Claire Mayer, Martine Côme, Lionel Ulmann, Graziella Chini Zittelli, Cecilia Faraloni, Hassan Nazih, Khadija Ouguerram, Benoît Chénais, Virginie Mimouni

**Affiliations:** 1EA 2160 MMS, Mer Molécules Santé, IUML FR 3473 CNRS, UFR Sciences et Techniques, 72085 Le Mans, CEDEX 9 and Institut Universitaire Technologique, Le Mans Université, 53020 Laval, CEDEX 9, France; claire.mayer@univ-lemans.fr (C.M.); martine.come@univ-lemans.fr (M.C.); lionel.ulmann@univ-lemans.fr (L.U.); benoit.chenais@univ-lemans.fr (B.C.); 2National Research Council, Department of Biology, Agriculture and Food Sciences, Tree and Timber Institute, 50019 Sesto Fiorentino (Florence), Italy; graziella.chinizittelli@cnr.it (G.C.Z.); faraloni@cnr.it (C.F.); 3EA 2160 MMS, Mer Molécules Santé, IUML FR 3473 CNRS, UFR Pharmacie, Université de Nantes, 44035 Nantes, CEDEX 1, France; el-hassane.nazih@univ-nantes.fr; 4UMR 1280 PhAN, Physiopathologie des Adaptations Nutritionnelles, CHU Hôtel Dieu, Université de Nantes, 44093 Nantes, CEDEX 1, France; khadija.ouguerram@univ-nantes.fr

**Keywords:** *Phaeodactylum tricornutum*, n-3 LC-PUFA, metabolic syndrome, dyslipidemia, inflammation

## Abstract

Long-chain polyunsaturated fatty acids, n-3 series (n-3 LC-PUFA), are known for their preventive effects against cardiovascular disease. In an unfavourable economic and environmental context of fish oil production, marine microalgae could be an alternative source of n-3 LC-PUFA and are of interest for human nutrition. The aim of this study was to evaluate the effects of *P. tricornutum*, a microalga rich in eicosapentaenoic acid and used as a food supplement, on the metabolic disorders associated with metabolic syndrome and obesity development. Three male Wistar rat groups (*n* = 6) were submitted for eight weeks to a standard diet or high-fat diet (HF) with 10% fructose in drinking water, supplemented or not with 12% of *P. tricornutum* (HF-Phaeo). Supplementation led to n-3 LC-PUFA enrichment of lipids in the liver, plasma and erythrocytes. Plasma transaminases showed no difference between the HF and HF-Phaeo groups. Body weight, fat mass, inflammatory markers and insulinemia decreased in HF-Phaeo rats versus the HF group. Plasma total cholesterol, triacylglycerols and leptine diminished in HF-Phaeo rats, while HDL-cholesterol increased. In conclusion, this study highlights the beneficial effects of *P. tricornutum* in reducing the metabolic disorders associated with metabolic syndrome.

## 1. Introduction

Metabolic syndrome (MS) is highly prevalent and associated with disturbances that include abdominal obesity, hypertension, dyslipidemia and hyperglycemia. MS is established with at least three criteria required for diagnosis: elevated waist circumference, elevated levels of triacylglycerol (TAG) and glucose in plasma, reduced plasma level of HDL-cholesterol (HDL-C), and high blood pressure. These disturbances represent a significant risk of developing cardiovascular diseases (CVD) [1]. Moreover, low-grade inflammation represents an important mechanism involved in the pathogenesis and development of obesity-related disorders and is also the link between adiposity, insulin resistance (IR), MS and CVD [2]. Moreover, non-alcoholic fatty liver disease (NAFLD) is defined as the main hepatic manifestation in MS, associated with IR and characterized by fat accumulation in hepatic tissue greater than 5% of liver weight. Other metabolic disturbances may be involved in the progression to more advanced stages of the disease, including oxidative stress and secretion of inflammatory mediators such as cytokines, adipokines and lipopolysaccharides, leading to the development and progression of inflammation, cell death, and fibrosis in non-alcoholic steatohepatitis (NASH) [3].

Several reports highlighted the positive role of long-chain polyunsaturated fatty acids n-3 series (n-3 LC-PUFA), mainly eicosapentaenoic acid (EPA) and docosahexaenoic acid (DHA), in the prevention of metabolic disorders associated with MS by their capacity to reduce the risk of developing CVD, cardiometabolic disorders as well as CVD-related mortality [4]. These beneficial effects are explained by the ability of these essential fatty acids (FA) to reduce plasma lipid levels such as TAG [5]. Moreover, it was demonstrated that n-3 LC-PUFA diet reduced plasma levels of the pro-inflammatory cytokines interleukin-6 (IL-6) and tumor necrosis factor-alpha (TNF-α), probably mediated by metabolic products of EPA and DHA with anti- inflammatory properties [5]. n-3 LC-PUFA such as EPA and DHA are poorly synthesized by humans from α-linolenic acid and the main dietary sources of these n-3 LC-PUFA are fish or algal oils. Thus, the European Food Safety Authority recommends an intake of 500 mg EPA/DHA per day, in the general healthy population, as a primary preventative measure against CVD. These amounts can be achieved by the ingestion of 1–2 fatty fish meals per week [6].

However, the decrease in halieutic resources and contamination concerns have made it necessary to have an alternative source of n-3 LC-PUFA [7]. Microalgae could be an interesting alternative and offer numerous advantages compared to fish oils. Indeed, microalgae constitute the first link in the food chain and therefore are less susceptible to contamination by heavy metals [8]. Moreover, unlike fish oils, the production of microalgae in controlled conditions does not induce any modification of the FA composition and allows for having constant n-3 LC-PUFA contents. Microalgae contain high lipid levels between 20% and 50% that are able to accumulate up to 80% of their dry weight in fat under stress conditions [8]. Finally, microalgae are a potential source of other highly bioactive molecules such as pigments and sterols, which are of interest to human nutrition [9]. Thus, microalgae seem to be an alternative for the prevention of the development of metabolic disorders associated with MS.

For centuries, microalgae have been consumed as a human food or as a dietary supplement because of their content in various macronutrients and micronutrients [10]. Nowadays, in the European Union (EU), the most popular microalgae widely commercialized as food ingredients are *Chlorella* and the procaryotic cyanobacterium *Arthrospira platensis*. Also, other microalgae are used as food in aquaculture, such as *Diacronema lutheri*, *Phaeodactylum tricornutum* and *Tisochrysis lutea*, but are not yet considered a food ingredient by the Novel Food Regulation of the EU [11].

*P. tricornutum* could be an alternative to fish oil due to its high content of protein, fiber, minerals and n-3 LC-PUFA, especially EPA [12]. *P. tricornutum* is rich in carotenoids, especially fucoxanthin, which is known to have anti-oxydative, anti-inflammatory, anti-dyslipidemia and anti-obesity effects [13]. To our knowledge, no nutritional study has highlighted the beneficial effects of *P. tricornutum* as a food supplement against MS installation.

The aim of this work was to evaluate the effects of the marine diatom *P. tricornutum*, used as a food supplement in the prevention of some disturbances associated with MS, such as body weight and adipose mass increases, dyslipidemia, inflammation and IR. Our nutritional experiment compared a high-fat (HF) diet, well known to induce MS risk factors, to the same HF diet supplemented with 12% of freeze-dried *P. tricornutum* [14]. The retained strain model is the male Wistar rat, which is commonly used to develop MS and adequately mimic all the aspects of human disease [15,16,17]. In order to reinforce the establishment of MS in rats, the HF diet was supplemented with 10% of fructose in drinking water. According to Panchal et al. [17], fructose is commonly used to induce MS in animal experiments and the increasing human consumption of fructose plays a major role in the obesity epidemic. In addition, fructose consumption is also involved in the development of IR and NAFLD [18]. The results of the present study showed the effects of *P. tricornutum* in reducing metabolic disorders induced by HF diet and fructose supplementation.

## 2. Materials and Methods

### 2.1. Animals and Diets

Eighteen male Wistar rats were obtained from Janvier Labs (Le Genest Saint Isle, France), aged three weeks and weighing 130 ± 10 g, to avoid age effects on metabolic disorders of MS. Male rats have been used to avoid any sexual endocrine disturbance. They were housed two per cage, 1291H Eurostandard Type III H in polycarbonate 425 × 266 × 185 mm (Tecniplast, Decines Charpieu, France) in a room under controlled conditions of temperature (22 ± 2 °C) and humidity (40–60%) and with a 12 h light/dark cycle. All animals were fed ad libitum with the standard diet A04 (SAFE, Augy, France) and had access to tap water for one week of acclimatization. The nutritional protocol and all the experiments have been approved by the Ethical Committee 06 Pays de la Loire and by the French Ministry of National Education, Higher Education and Research (procedure APAFIS 6737, 20 October 2016).

After a week of acclimatization, the animals were randomly divided into three groups of six rats and were assigned to receive diets ad libitum for eight weeks as follows: (1) the control (CTRL) group, continued to receive the standard diet A04 providing 3.35 kcal/g, 72 kcal%, 19 kcal%, 8 kcal% from carbohydrates, proteins and lipids, respectively; (2) the HF group was fed the 260 HF high-fat diet (Safe, Augy, France) with 10% fructose in ad libitum drinking water (DW) (providing 1.67 kcal/mL) (Distriborg, St. Genis-Laval, France). HF diet provided 22 kcal/g, 61 kcal%, 24 kcal% from fat and carbohydrates, respectively; (3) the HF-Phaeo group received a HF diet supplemented with 12% (*w*/*w*) of freeze-dried microalga *P. tricornutum* (CNR, Florence, Italy). The microalga supplementation with the dose of 12% was chosen on the basis of previous studies that showed the beneficial effects of the marine microalga *O. aurita* at 12% after eight weeks of the diet [19,20]. This microalga, which is rich in EPA, prevents metabolic disorders associated with CVD induced by a high-fat diet in Wistar rats [19,20]. Moreover, the chosen dose of fructose supplementation was based on a meta-analysis highlighting that a dose of 10% fructose in drinking water was sufficient to induce the first characteristics of MS such as an increase in body weight, blood pressure and glucose, insulin and triglyceride plasma levels in rats [21]. These data have been confirmed by a previous study [14]. Freeze-dried *P. tricornutum* (Phaeo) provided 1.67 kcal/g, 38 kcal%, 35 kcal%, 22 kcal% from crude protein, lipids and carbohydrates, respectively. Microalgal supplementation was incorporated directly in the HF diet to create a homogeneous mixture and provide 0.48 kcal/g in the HF-Phaeo diet. EPA content in *P. tricornutum* was 2.04% of dry matter, equivalent to an EPA averaged intake of 33 mg/day/rat. HF diet was stored at + 4 °C and Phaeo at −20 °C and renewed in the cages every three days for eight weeks.

The daily food and water consumption was evaluated in order to calculate energy intake. Energy intake (kcal/day) means food consumption × dietary metabolizable energy. The body weight gains of the rats were monitored three times a week. Daily food and water consumption, and energy intake, were reported according to body weight.

The main characteristics and the FA, pigment and sterol compositions of the CTRL diet, HF diet and *P. tricornutum* biomass are reported in Appendix A.

The main components of the CTRL diet, HF diet and *P. tricornutum* are reported in Appendix A. Data from CTRL and HF diets were provided by SAFE (Augy, France). Concerning *P. tricornutum*, biomass was analysed for protein, carbohydrate, lipid, dietary fiber, ash, and moisture. Total protein content was estimated as N × 6.25, where N is the nitrogen content determined through the elemental analysis. Carbohydrate was determined following Dubois et al. [22], and lipids following Marsh and Weinstein [23]. Total dietary fiber (TDF), insoluble dietary fiber (IDF) and soluble dietary fiber (SDF) were determined by AOAC Method 985.29 (AOAC Official Method 985.29) using commercial kits (Megazyme, Bray, Ireland). TDF was experimentally analysed, not calculated as a sum of SDF and IDF. Moisture and ash were analysed following ISTISAN protocols (ISTISAN Report 1996/34, method B, p. 7; ISTISAN Report 1996/34, pp. 77–78, respectively). The FA compositions of the CTRL diet, HF diet and *P. tricornutum* biomass are presented in Appendix A. The determination of the FA composition of the CTRL diet is described below. For the HF diet, the FA composition was determined according to Simonato et al. [24]. The FA analysis of *P. tricornutum* was performed according to the ISO 12966-4:2015 + ISO 12966-2:2011 procedures.

Pigment and sterol composition, in vitro digestibility, and antioxidant activity of *P. tricornutum* are reported in Appendix A. Pigment composition was performed by the SCOR-UNESCO method [25]. Carotenoid content was determinated by HPLC analysis according to Van Heukelem and Thomas [26]. In vitro digestibility was evaluated by the method of Boisen and Fernández (1997), modified as reported by Batista et al. [27,28].

Antioxidant activity of extracts in 90% acetone was measured by using the 2,2-diphenyl-1-picrylhydrazyle (DPPH) radical scavenger according to the method of Bondet et al. with slight adaptations, reported in Appendix A [29].

### 2.2. Blood and Organ Sampling

After the eight weeks of the nutritional protocol, all rats were fasted for 12 h and anesthetized by intraperitoneal administration of a Diazepam–Ketamine mix (4:3, *v*/*v*). Blood was collected from the abdominal aorta and sampled in 10% ethylenediaminetetraacetic acid (EDTA) (from Sigma, St. Louis, MO, USA) coated tubes. Total blood was centrifuged at 1000× *g* for 10 min; the supernatant containing the plasma fraction was aliquoted in polyethylene tubes and the pellet of red blood cells (RBC) was separately collected. Then, plasma and RBC samples were stored at −20 °C. Liver, epididymal and abdominal adipose tissues were removed, rinsed with ice-cold NaCl solution (0.9%), weighed, frozen in liquid nitrogen, and stored at −80 °C until analysis.

### 2.3. Biochemical Plasma Analyses

Plasma levels of glucose, TAG, total cholesterol (TC), HDL-C, aspartate amino-transferase (ASAT) and alanine amino-transferase (ALAT) were measured by enzymatic methods using commercial enzyme kits (BIOLABO, Maizy, France). From ASAT and ALAT measures, ASAT/ALAT ratio was calculated. The atherogenic index of plasma (AIP) was calculated as log (TAG/HDL-C) [30]. Pro-inflammatory cytokines, including IL-6, TNF-α and interleukin-4 (IL-4), and anti-inflammatory cytokine interleukin-10 (IL-10), as well as leptin, were quantified using rat enzyme-linked immunosorbent assay kits (ELISA) from Abcam (Cambridge, UK) according to the manufacturer’s protocols. The insulin level was evaluated using ELISA kit from Thermo Scientific (Waltham, MA, USA). The homeostasis model assessment of insulin resistance index (HOMA-IR) was estimated by calculating the fasting plasma glucose concentration (mg/dL) multiplied by fasting insulinemia (µUI/mL), divided by 405 [31].

### 2.4. Fatty Acid Composition Analyses

FA profiles in rat liver, plasma and RBC were determined by gas chromatography–flame ionization detection (GC-FID). Briefly, total lipids were extracted from the liver with choloroforme/methanol (2:1, *v*/*v*) according to the method of Folch et al. [32]. Furthermore, total lipids were extracted from the plasma and RBC with methylal/methanol (4:1, *v*/*v*) according to the method of Delsal [33]. Phospholipids and neutral lipids were separated from the total lipids by solid phase extraction using silica gel columns Sep-pack (Sep-pack plus, silica cartridges, Waters, Guyancourt, France). Fatty acid methyl esters (FAMEs) were obtained according to the method of Slover and Lanza [34], and analysed using a FOCUS gas chromatography instrument (Thermo Electron Corporation, Les Ulis, France) equipped with a capillary column CP Sil-88 25 m × 0.25 mm (Varian, Les Ulis, France). Then, FAMEs were detected with a flame-ionization detector. Each FA was identified from an authentic fatty acid methyl ester standard (Sigma-Aldrich, Saint-Quentin Fallavier, France) and results were expressed as a molar percentage (mol %).

### 2.5. Hepatic Lipid Measurements

From an aliquot of the total lipid extract from liver, cholesterol and TAG levels were determined by enzymatic methods using commercial enzyme kits (BIOLABO, Maizy, France).

### 2.6. Statistical Analysis

Data from experimental analyses are presented as mean values ± standard deviation (SD) (*n* = 6). After the analysis of variance by one-way ANOVA, the mean values were compared using Fisher’s least significant difference post hoc test (LSD). All statistical analyses were performed with Statgraphics Plus 5.1 (Manugistics Inc., Rockville, MD, USA).

## 3. Results

### 3.1. Effects of P. tricornutum on Body and Organ Weight

#### 3.1.1. Food and Water Intake

Food and water intake in the experimental groups were monitored for eight weeks (Appendix A). The CTRL group displayed the highest food consumption relative to the body weight during the experimental period (ANOVA, *p* < 0.001) when compared with the other groups. The HF and HF-Phaeo groups presented similar food consumption/body weight ratios over time. At eight weeks, the ratio of water intake/body weight (with 10% fructose in the HF and HF-Phaeo diets) was markedly higher in rats fed an HF diet than in the CTRL group (ANOVA, *p* < 0.001), except for in the first week (ANOVA, *p* < 0.01). Although the water intake/body weight ratio of HF-Phaeo rats was lower than that of the HF group after the fourth week, it was higher than the CTRL group, except at week 8 (ANOVA, *p* < 0.01, ANOVA, *p* < 0.001, for weeks 1, 4 and 8, respectively).

#### 3.1.2. Energy Intake

Energy intake was calculated from water and food consumption, relative to body weight, and was higher for yjr HF-Phaeo group throughout the protocol (ANOVA, *p* < 0.001) (Appendix A). The energy intake of HF rats was higher than that of the CTRL group and statistically different from the fourth week of treatment (ANOVA, *p* < 0.001) and similar to the first week of the protocol.

#### 3.1.3. Body and Organ Weights

Despite a similar energy intake between HF rats and those fed with *P. tricornutum* at week 8, the body weight of these two animal groups was significantly different. Indeed, after two months of treatment, the final body weight was higher in the HF group compared to the other groups (ANOVA, *p* < 0.01), and the HF-Phaeo group body weight was similar to control rats (Table 1). Abdominal and epididymal adipose tissues as well as liver tissue weights increased with the HF diet compared with the other groups (ANOVA, *p* < 0.001 for abdominal and epididymal adipose tissues, *p* < 0.05 for liver tissue). Supplementation with *P. tricornutum* significantly reduced the abdominal adipose tissue weight compared to the HF group (ANOVA, *p* < 0.001) and maintained the weight of epididymal adipose tissue and liver tissue at the same level as in the control rats.

### 3.2. Effects of P. tricornutum on Fatty Acid Composition of Plasma, RBC and Liver Lipids

#### 3.2.1. SFA Contents

Rats fed the HF diet showed a higher content of 14:0 in plasma total lipids (Table 2, ANOVA, *p* < 0.01), phospholipids, total lipids and neutral lipids of liver (Table 4, Appendix A, ANOVA, *p* < 0.001), with an increase in SFA level in liver neutral lipids (Appendix A, ANOVA, *p* < 0.01) compared to other groups. In contrast, a decrease of the SFA content in total lipids of RBC was observed in HF rats compared to the CTRL group (Table 3, Appendix A, ANOVA, *p* < 0.01).

#### 3.2.2. n-6 LC-PUFA Contents

The level of n-6 LC-PUFA was significantly higher in phospholipids and total lipids of RBC of HF rats compared to other groups (Table 3, Appendix A, ANOVA, *p* < 0.001), particularly for 20:4n-6 (ANOVA, *p* < 0.01 and *p* < 0.001 for phospholipids and total lipids, respectively). However, a decrease of n-6 LC-PUFA level, especially for 18:2n-6 and 20:4n-6, was observed in plasma total lipids and liver neutral lipids of HF rats compared to the CTRL group (Table 2, ANOVA, *p* < 0.01 and *p* < 0.001; Appendix A, ANOVA, *p* < 0.001 and *p* < 0.01). Moreover, in the HF group, data also showed a significant decrease of n-6 LC-PUFA level in liver phospholipids (Table 4, ANOVA, *p* < 0.001) as well as of the 18:2n-6 level in liver total lipids (Appendix A, ANOVA, *p* < 0.001) in comparaison with CTRL rats. In liver total lipids, the level of n-6 LC-PUFA, including 20:4n-6, decreased in the HF group compared to the other two groups (Appendix A, ANOVA, *p* < 0.001).

#### 3.2.3. n-3 LC-PUFA contents

n-3 LC-PUFA contents increased with *P. tricornutum* supplementation in plasma total lipids (Table 2, ANOVA, *p* < 0.05), total lipids and phospholipids of RBC (Appendix A, ANOVA, *p* < 0.01; Table 3, ANOVA, *p* < 0.001), total lipids, neutral lipids and phospholipids of liver (Appendix A and Table 4, ANOVA, *p* < 0.001), especially for 20:5n-3 (Table 2, Table 3 and Table 4, Appendix A, *p* < 0.001) when compared to the other experimental groups. The 22:5n-3 content was higher in total lipids of plasma (Table 2, ANOVA, *p* < 0.001 and *p* < 0.01), phospholipids of RBC (Table 3, ANOVA, *p* < 0.001), total lipids, neutral lipids and phospholipids of liver (Table 4, Appendix A, ANOVA, *p* < 0.001) from the HF-Phaeo group compared to the other groups. Moreover, an increase of 22:5n-3 level in total lipids of RBC was observed in HF-Phaeo rats compared to HF rats (Appendix A, ANOVA, *p* < 0.05). This increase of n-3 LC-PUFA in lipids of the HF-Phaeo rats was associated with a significant decrease in the n-6 PUFA/ n-3 PUFA ratio (ANOVA, *p* < 0.001).

#### 3.2.4. MUFA Levels

The FA composition of plasma total lipids revealed an increase in MUFA levels with the HF diet, particularly for 18:1(n-7 + n-9), compared to the CTRL group (Table 2, ANOVA, *p* < 0.01). A similar trend was noticed with 18:1(n-7 + n-9) levels in total lipids and phospholipids of liver, phospholipids and total lipids of RBC of HF rats (Table 3, Appendix A, ANOVA, *p* < 0.05) (Appendix A, ANOVA, *p* < 0.001; Table 4, ANOVA, *p* < 0.05). Plasma total lipids and liver phospholipids of the HF-Phaeo group displayed lower MUFA levels, especially for 18:1(n-7 + n-9), compared to HF rats (Table 2, Table 4, ANOVA, *p* < 0.01). The same trend was noticed for 18:1(n-7 + n-9) contents in liver neutral lipids (Appendix A, ANOVA, *p* < 0.01).

#### 3.2.5. Δ9-Desaturase Index

The Δ9-Desaturase index was calculated from the content in 18:0 and 18:1(n-7 + n-9) as follows: [18:1(n-7 + n-9)/ 18:0 + 18:1(n-7 + n-9)] and showed no difference between the groups except for a significant decrease in liver neutral lipids of HF rats compared to the other groups (Appendix A, ANOVA, *p* < 0.05). With *P. tricornutum* supplementation, the Δ9-Desaturase index was similar to that of CTRL rats in liver neutral lipids and lower than that of the HF group in total lipids of red blood cells (Appendix A, ANOVA, *p* < 0.05).

### 3.3. Effects of P. tricornutum Supplementation on Physiological and Metabolic Disorders in Wistar Rats Fed a High-Fat Diet

#### 3.3.1. Plasma Lipid Levels and AIP Index

In the HF group, basal plasma glucose level was higher compared to control group (ANOVA, *p* < 0.05) and plasma levels of insulin and leptin were higher than in rats fed with other diets (ANOVA, *p* < 0.001) (Table 5). Supplementation with *P. tricornutum* partially prevented the increase of insulin level observed in the HF group and restored the basal leptinemia level, whereas glycemia was not restored in the HF-Phaeo group. In accordance with these results, the HOMA-IR index increased with the HF diet (ANOVA, *p* < 0.001) and was restored in the HF-Phaeo group (Table 5). The HF group showed higher plasma TC and TAG levels compared to the CTRL group (ANOVA, *p* < 0.01 and *p* < 0.001) (Figure 1a,b). Supplementation with *P. tricornutum* decreased triglyceridemia and restored cholesterolemia to the control level. The HF-Phaeo group evidenced an increase in HDL-C level (Figure 1c) compared to the other groups (ANOVA, *p* < 0.001). However, no difference was noticed between HF and control rats. These results were associated with a high AIP, an effective index for estimating abdominal obesity, observed in the HF group (ANOVA, *p* < 0.001) (Table 5). By contrast, a marked decrease of AIP was observed with *P. tricornutum* supplementation compared to the CTRL and HF groups (ANOVA, *p* < 0.001) (Table 5).

#### 3.3.2. Transaminase and Hepatic Lipid Levels

We also studied the integrity and metabolic function of the liver. In the rats fed HF and HF-Phaeo diets, the plasma level of ASAT increased and the plasma ALAT level as well as the ASAT/ALAT ratio decreased significantly (ANOVA, *p* < 0.001) compared to the CTRL group (Table 5). Finally, in the liver, the HF diet induced a significant increase of TAG and TC levels, while *P. tricornutum* supplementation prevented hepatic accumulation of these lipids (Table 5).

### 3.4. Effects of P. tricornutum on Inflammatory Status

#### 3.4.1. Pro-Inflammatory Cytokines

As shown in Figure 2, plasma concentrations of pro-inflammatory cytokines, including TNF-α and IL-6, were significantly enhanced in the HF group (ANOVA, *p* < 0.001) compared to those in the CTRL and HF-Phaeo groups. The results also evidenced that *P. tricornutum* supplementation restored IL-6 concentration and significantly decreased TNF-α concentration in the plasma compared to CTRL group (ANOVA, *p* < 0.001).

#### 3.4.2. Anti-Inflammatory Cytokines

Concerning anti-inflammatory cytokines, IL-4 concentration in plasma and IL-10 concentration in adipose tissue decreased with the obesogenic diet compared to standard diet (ANOVA, *p* < 0.001). Supplementation with *P. tricornutum* significantly improved inflammatory status by an increase of IL-4 and IL-10 levels in the plasma and adipose tissue, respectively, compared with the HF group (ANOVA, *p* < 0.05). Nevertheless, plasma and adipose tissue concentrations of IL-4 and IL-10 in the HF-Phaeo group were not completely recovered compared to control rats (ANOVA, *p* < 0.001).

## 4. Discussion

The aim of this work was to study the impact of *P. tricornutum* used as a food supplement on metabolic disorders associated with the progression to MS, including overweight, dyslipidemia, IR and inflammation. The results showed that *P. tricornutum* supplementation led to an enrichment in n-3 LC-PUFA in tissues. *P. tricornutum* supplementation exerted beneficial effects on body weight loss and reduction in adipose tissue weight reduction. It also contributed to an improvement of plasma lipid parameters, insulinemia, leptinemia and inflammatory status.

The results showed higher levels of n-3 LC-PUFA in the FA composition of plasma, RBC and liver lipids in rats fed an HF diet supplemented with *P. tricornutum*. These results are in agreement with those described in another study using the diatom *O. aurita*, rich in EPA, used as a food supplement in a rat model [19]. This suggests that nutritional treatment has been effective and that the tissue enrichment in n-3 LC-PUFA could explain the results described below.

The HF diet is well known to induce obesity and metabolic disorders similar to MS in humans, especially by an increase in body weight [35]. In the present study, despite a higher energy intake in the HF-Phaeo rats compared to the other groups, our results showed that HF-Phaeo body weight was lower than that observed in the CTRL and HF groups. Similar results were obtained with microalgal oil from a mutant derived from *Thraustochytriidae* sp., which contains more than 50% n-3 LC-PUFA in obese mice after nine weeks of treatment [36]. The high EPA content of *P. tricornutum* (2.04% of dry weight), which is equivalent to a daily intake of 33 mg/rat, could explain the decrease in body weight and fat mass in the HF-Phaeo group. In a rodent study, EPA supplementation, at a dose of 36 g/kg of body weight, prevented the weight gain induced by the HF diet and fructose and reduced epididymal adipose tissue mass after 10 weeks of the diet [37]. These observations could be explained by a limiting effect of EPA intake on both hypertrophy and hyperplasia of fat cells attributed to upregulation of the adipose peroxisome proliferator-activated receptor gamma (PPARγ) gene, which is involved in adipocyte differentiation but also in the increase of fat deposits and apoptosis of adipose tissue [38,39]. In addition, the decrease of body weight and fat mass in HF-Phaeo rats could also be explained by the high level of fucoxanthin found in the biomass of *P. tricornutum* (62.3% of carotenoids, 34% of total pigments and 0.31% of dry weight), which represents a daily intake of fucoxanthin of about 5 mg/rat, and agrees with the data obtained in mice supplemented with a lipid fraction of *Undaria pinnatifida*, which contains 9.6% fucoxanthin [40]. The reduction of weight gain and white adipose tissue accumulation via fucoxanthin could be mediated by the increase of uncoupling protein 1 (UCP1) expression in white adipose tissue [40]. UCP1 is mainly expressed in brown adipose tissue and acts in thermogenesis, energy expenditure regulation, and protection against oxidative stress [41]. Another study showed that synthetic fucoxanthin or *Undaria pinnatifida* supplementation (400 mg/kg of body weight), which is rich in fucoxanthin (0.98 mg/g of dry weight), improved energy expenditure, β-oxidation and adipogenesis by upregulating gene expression of PPARα, PGC1α, PPARγ and UCP-1 in rats fed with a hyperlipidic diet. These data could explain the lower body weight and fat mass observed in HF-Phaeo rats, while the energy intake/body weight ratio was higher than that of other groups [42]. Dietary fiber is also known to exert positive effects on body weight reduction, and the potential impact of *P. tricornutum* dietary fiber cannot be excluded. The effects of dietary fiber on body weight regulation could be due to the increase of intraluminal viscosity and fermentation of short-chain fatty acids. These physiological changes could lead to a decrease in food intake, enhanced by satiation and/or satiety. Dietary fiber also decreases gastric emptying, slows energy and nutrient absorption, and may also influence fat oxidation and fat storage [43].

Although C 18:1(n-7 + n-9) levels were increased in the plasma, RBC and liver tissues of rats fed with a HF diet and fructose, the Δ9-Desaturase or stearoyl-CoA desaturase (SCD1) index, which reflects the levels of endogenously synthesized FA in the liver, was not signicantly higher than that of the CTRL and HF-Phaeo groups. These results suggest that the increase of MUFA may be a consequence of the increase of lipogenic gene expression other than SCD1. Indeed, a fructose-enriched diet contributes to the risk of MS, IR and obesity, mediated by the activation of the carbohydrate-responsive element-binding protein (ChREBP) transcription factor in synergy with sterol regulatory element-binding proteins-1C (SREBP-1c) to increase the expression of lipogenic genes such as acetyl CoA carboxylase (ACC) and fatty acid synthase 1 (FAS1) [44]. Otherwise, our study showed a decrease of n-6 LC-PUFA levels in total lipids of plasma and phospholipids of liver of HF rats. These changes could be explained by mechanisms other than desaturase/elongase activity and could be dependent on other aspects of lipid metabolism including oxidation, substrate biodisponibility, acylation, second messenger synthesis, nutritional and hormonal status [45]. Moreover, the decrease of 18:2n-6 (linoleic acid), which is a precursor for the synthesis of 20:4n-6 (arachidonic acid), could explain the decrease of arachidonic acid observed in total lipids of plasma and phospholipids of liver in the HF group [46].

Significant n-3 LC-PUFA incorporation was observed in all tissues, particularly EPA and docosapentaenoic acid (DPA), but not DHA, suggesting that the conversion from EPA to DHA is limited by the final conversion from DPA to DHA [47]. DPA can also contribute to the beneficial effects of EPA supplementation on fat mass, dyslipidemia and inflammation associated with MS and obesity [48]. Otherwise, MUFA levels were decreased in liver phospholipids of rats fed with a HF-Phaeo diet, probably due to the decrease of Δ9-Desaturase activity. This result suggests that the HF-Phaeo diet, rich in EPA, modulated the FA composition of the liver. A previous report demonstrated that the incorporation of n-3 LC-PUFA into hepatic phospholipids decreased the SCD1 activity and led to a hepatic FA profile change via reduced desaturase activities [49].

Dyslipidemia is a component of MS, and we showed that the HF group rats exhibited higher plasma TC and TAG levels, although no difference was noticed between HF and control rats in terms of HDL-C plasma levels [50]. In the present study, the effects of the entire biomass of *P. tricornutum* were investigated, showing that, when used as a food supplement, it reduced dyslipidemia. The diatom *P. tricornutum* contains large amounts of EPA, as well as other nutritional substances such as fiber, phytosterols and fucoxanthin. These bioactive molecules have beneficial effects on the regulation of lipid metabolism [51,52,53]. First, phytosterols are present in the HF-Phaeo diet (0.65% of dry weight), equivalent to a phytosterol intake of about 10.5 mg/day/rat. Phytosterols have a structure similar to that of cholesterol and can replace them in intestinal absorption, leading to an increase in cholesterol excretion by bile salts and contributing to a decrease of plasma cholesterol levels and especially LDL-cholesterol (LDL-C) [51]. Secondly, an anti-dyslipidemic effect has been shown for polysaccharides from the red microalga *Porphyridium* (27% insoluble and 8.5% soluble dietary fiberon a dry weight); this caused a lower LDL-C level in Sprague–Dawley rats [54]. Some of these microalgal polysaccharides are considered natural dietary fiber, recognized for the beneficial effects on intestinal transit; they also contribute to maintaining cholesterolemia at a basal level and therefore could prevent CVD [55]. Thus, the dietary fiber found in *P. tricornutum* (6.29% insoluble and 6.06% soluble dietary fiber on a dry weight basis) could also play a role in the decrease of dyslipidemia in HF-Phaeo rats, by a daily intake per rat of about 181 mg of total dietary fiber, including 102 mg and 98 mg of insoluble and soluble dietary fiber, respectively. Finally, EPA has a positive effect on the regulation of lipid metabolism and the prevention of dyslipidemia and related diseases. The hypotriglyceridemic effect of n-3 LC-PUFA could then be explained by the inhibition of hepatic lipogenesis and TAG secretion [56].

Dyslipidemia is the most important risk factor for atherosclerosis [57], and the increase of AIP is considered a major predictor of atherosclerosis and an effective index for estimating abdominal obesity [58]. The atherogenic effect of the HF diet is well described [59], and, in agreement with this, our study showed that the HF diet promoted abdominal obesity and a marked increase of AIP, while supplementation with *P. tricornutum* improved the lipid profile and decreased AIP. These effects may be related to the various molecules present in *P. tricornutum* such as n-3 LC-PUFA, phytosterols and fiber, all of them recognized as having cardiovascular benefits [4,60,61]. Moreover, according to previous studies, there is a relationship between the intake of n-3 LC-PUFA and the incorporation of n-3 LC-PUFA into the RBC of the HF-Phaeo group, which can be considered a prevention marker for CVD [62,63]. Supporting this, a relationship between the consumption of food enriched with n-3 LC-PUFA, the increase of n-3 LC-PUFA in RBC and the reduction of cardiovascular risk has been highlighted [63]. In addition, there would be a relationship between n-3 LC-PUFA content in RBC, named omega 3-index, and obesity. A correlation between erythrocyte enrichment in n-3 LC-PUFA and the decrease of body mass index, waist circumference and body fat has already been demonstrated [64].

A marked reduction of TAG and cholesterol levels was previously reported in Wistar rats fed marine microalga *Diacronema vlkianum* biomass for 66 days, equivalent to 101 mg/kg of diet in EPA + DHA, suggesting that n-3 LC-PUFA from MA modulate the FA composition of liver and prevent hepatic steatosis installation [65]. This may be mediated by the decrease of gene expression of fatty acid synthase (FAS) and sterol regulatory element-binding protein 1-c (SREBP1-c), which are involved in lipid synthesis [65,66]. The liver is not a lipid storage organ and an excess of TAG and cholesterol can be hepatotoxic [67]. In accordance with a previous study showing a decrease of ASAT/ALAT ratio in young Wistar rats fed a fructose-enriched diet, these results suggest a liver injury due to fructose supplementation [67]. Indeed, fructose, in contrast to glucose, does not lead to insulin secretion by β pancreatic cells, probably because of the absence of the fructose transporter at the surface of the Langerhans islets beta cells. Fructose is drastically absorbed by the liver via glucose transporter 2 (GLUT2) and then metabolized to produce glucose, glycogen, pyruvate, lactate, glycerol and acyl glycerol molecules. The lipogenic properties of fructose result in the accumulation of triglycerides and cholesterol, which leads to IR and glucose intolerance [14,16].

It can be noticed that the ASAT/ALAT ratio is mainly modified by the increase of ALAT levels in the liver, a specific hepatotoxicity biomarker in NAFLD, although the ASAT level was not altered by the HF diet with 10% fructose. Supplementation with *P. tricornutum* did not decrease hepatotoxicity induced by HF diet, but it did not amplify this metabolic disorder, suggesting that *P. tricornutum* supplementation, at the dose of 12%, was not hepatotoxic when incorporated in the HF diet.

Adipose tissue serves not only as energy storage but also has an endocrine function, secreting and releasing various mediators (adipokines, cytokines) that may have pro- or anti-inflammatory activities. The present study showed that the HF diet induced an increase of TNF-α and IL-6 pro-inflammatory cytokines plasma level and a decrease of IL-4 and IL-10 anti-inflammatory cytokines plasma level, associated with hyperinsulinemia and a high HOMA-IR index. By contrast, the plasma level of TNF-α and IL-6 pro-inflammatory cytokines, insulinemia and HOMA-IR index were decreased in HF-Phaeo rats. In addition, IL-4 and IL-10 levels were increased in the plasma and adipose tissue of the HF-Phaeo group, reflecting the partial restoration of the basal inflammatory status in rats supplemented with *P. tricornutum*. Thus, *P. tricornutum* could play a preventive role against inflammation and insulinoresistance, as observed in a previous study that used the EPA-rich microalga *O. aurita* as a food supplement in the HF diet [68]. The restoration of inflammatory status and insulinemia could be due to various molecules contained in *P. tricornutum* such as the carotenoid pigments, which are known anti-inflammatory agents [69]. In a recent study, fucoxanthin, at a dose of 0.6% during four weeks, significantly inhibited obesity by the decrease of inflammatory markers production such as TNF-α and cyclo-oxygenase-2 (COX-2) in HF-fed mice [70]. In addition, fucoxanthin, soluble dietary fiber and phytosterols present in *P. tricornutum* could exert significant anti-inflammatory activity by the decrease of pro-inflammatory cytokines plasma levels observed in HF-Phaeo rats. Moreover, marine microlgae contain a large diversity of n-3 LC-PUFA, known for their inhibitory effects on inflammation [71]. Our results suggest that n-3 LC-PUFA, and specifically EPA, which is abundant in *P. tricornutum*, would decrease pro-inflammatory cytokine secretion and increase anti-inflammatory cytokine production, thus reducing low-grade inflammation and preventing hyperinsulinemia. In agreement, a previous study showed that lipid extracts of *D. lutheri*, a microalga rich in EPA and DHA, had an anti-inflammatory effect on the human macrophage line THP-1 by inhibiting the production of pro-inflammatory cytokines and the transcription of genes involved in inflammatory signaling pathways [72]. In our study, the low inflammation observed with *P. triconutum* supplementation may be due to the synergistic effect of various bioactive molecules such as pigments and FA. It seems more appropriate to consider the synergistic effects between various bioactive molecules in the prevention of inflammation associated with obesity than to consider the effect of individual molecules. Further research is needed to investigate the involvement of *P. tricornutum* molecules in the modulation of inflammation and IR signaling pathways in the liver and adipose tissue.

Hyperleptinemia is considered a marker of pro-inflammatory status, positively correlated with body fat, and leptin is a mediator of inflammation in obese adults [73]. In the present study, HF diet induced hyperleptinemia, while supplementation with *P. tricornutum* decreased the leptin level. This is consistent with the results of Yook et al., who showed a significant decrease in leptin plasma concentration, after eight weeks of treatment, in C57BL/6J mice fed a HF diet supplemented with *Aurantiochytrium* microalgal oil as a source of n-3 LC-PUFA [74]. Thus, n-3 LC-PUFA could be a potential agent against hyperleptinemia. Carotenoids, and especially fucoxanthin, are other bioactive molecules from *P. tricornutum*, which could be involved in leptinemia regulation. Indeed, a study highlighted the decrease of insulin and leptin plasma levels in C57BL/6J mice fed a HF diet containing *P. tricornutum* extract (corresponding to 0.2% of fucoxanthin) for eight weeks [75].

## 5. Conclusions

In conclusion, we have shown that *P. tricornutum* might be a promising marine source of novel food ingredients for the prevention of MS and obesity. Supplementation with this microalga leads to effective assimilation of its constituents, in particular lipids, as attested by membrane enrichment with n-3 LC-PUFA. Feeding rats a *P. tricornutum* supplement resulted in the reduction of body weight and adipose tissue weight and lower liver and plasma lipid levels, and tended to prevent insulin resistance and CVD. The bioactivity exerted by *P. tricornutum* extracts on inflammation and IR signaling pathways involved in human hepatocyte lipotoxicity should be further explored in vitro. Another interesting experiment would be to pair *P. tricornutum* extracts with other nutraceutical compounds to study their impact in fructose-induced NAFLD. Indeed, nutraceuticals containing microalgae, alone or in association with other plant extracts, are used to antagonize the effects from agents causing hepatic inflammation and prevent liver injury [76]. *P. tricornutum* extracts, which are rich in fucoxanthin, reduced liver lipid accumulation in mice [13]. Anthocyanins or phenolic acids are nutraceutical coupounds known to exert hepatoprotective and hepatotropic effects and, associated with *P. tricornutum* extracts, could have greater efficicency in the prevention of fructose-induced NAFLD [77,78].

## Figures and Tables

**Figure 1 nutrients-11-01069-f001:**
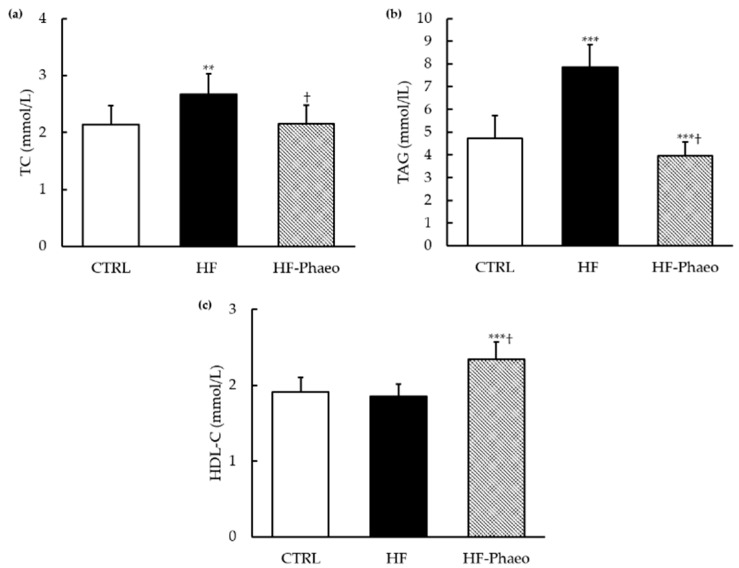
TC (**a**), TAG (**b**) and HDL-C (**c**) of the control animals, HF animals and HF animals that received *P. tricornutum*. CTRL, control group; HF, high-fat group; HF-Phaeo, high-fat group supplemented with *P. tricornutum*; HDL-C, high-density lipoprotein; TC, total cholesterol; TAG, triacylglycerol. Values are means (*n* = 6), with standard deviations represented by vertical bars. Mean values were significantly different from those of the CTRL group: ** *p* < 0.001; *** *p* < 0.0001 (one-way ANOVA; LSD post hoc test). ^†^ Significant difference compared with the HF group (*p* < 0.05).

**Figure 2 nutrients-11-01069-f002:**
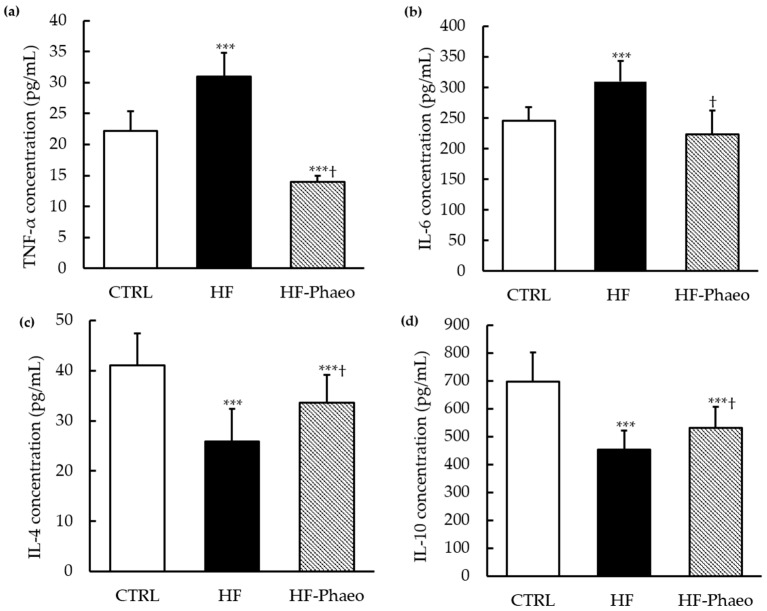
Effects of *P. tricornutum* supplementation on inflammatory biomarkers. The plasma concentrations of (**a**) TNF-α, (**b**) IL-6, (**c**) IL-4 and adipocyte concentration in (**d**) IL-10. CTRL, control group; HF, high-fat group; HF-Phaeo, high-fat group supplemented with *P. tricornutum*; IL-4, interleukin 4; IL-6, interleukin 6; IL-10, interleukin 10; TNF-α, tumor necrosis factor-α. Values are means (*n* = 6), with standard deviations represented by vertical bars. Mean values were significantly different from those of the CTRL group: *** *p* < 0.0001 (one-way ANOVA; LSD post hoc test). ^†^ Significant difference compared with the HF group (*p* < 0.05).

**Table 1 nutrients-11-01069-t001:** Animal characteristics.

	CTRL		HF		HF-Phaeo	
Parameter	Mean	SD	Mean	SD	Mean	SD
Final BW at day 56 of treatment (g)	457.86	25.32	512.75 **	28.82	468.75 ^†^	38.37
LW/BW (%)	2.57	0.20	2.76 *	0.16	2.62	0.13
AAT/BW (%)	1.61	0.27	4.12 ***	0.32	2.26 ***^†^	0.51
EAT/BW (%)	1.63	0.20	2.76 ***	0.14	1.86	0.36

AAT, abdominal adipose tissue weight; BW, body weight; CTRL, standard diet; EAT, epididymal adipose tissue weight; HF, high-fat diet; HF-Phaeo, high-fat diet supplemented with 12% of *P. tricornutum*; LW, liver weight. Results are represented as mean values ± SD, *n* = 6. Mean values were significantly different from those of the CTRL group: * *p* < 0.05; ** *p* < 0.001; *** *p* < 0.0001 (one-way ANOVA; LSD post hoc test). ^†^ Significant difference compared with the HF group (*p* < 0.05).

**Table 2 nutrients-11-01069-t002:** Fatty acid composition of plasma total lipids.

	CTRL		HF		HF-Phaeo	
Fatty Acids (mol %)	Mean	SD	Mean	SD	Mean	SD
**SFA**						
14:0	0.47	0.05	1.35 **	0.31	0.88 **^†^	0.18
16:0	22.07	2.95	22.21	1.84	17.51 *^†^	2.27
18:0	9.52	1.76	11.51	0.60	10.68	1.80
Total SFA	28.97	1.89	35.07	3.90	29.07	4.21
**MUFA**						
16:1	2.27	0.28	2.10	0.60	1.86	0.24
18:1 (*n*-7 + *n*-9)	13.33	1.75	18.80 **	3.41	12.91 ^†^	0.30
20:1*n-9*	ND	-	ND	-	ND	-
22:1	ND	-	ND	-	ND	-
24:1*n-9*	ND	-	ND	-	ND	-
Total MUFA	15.60	2.01	20.90 *	3.96	14.90 ^†^	0.52
**PUFA**						
18:2*n*-6	16.44	1.32	10.69 ***	1.42	10.38 ***	0.91
20:2*n*-6	0.44	0.11	1.26 **	0.18	0.74 ^†^	0.23
20:4*n*-6	22.18	1.56	17.20 **	2.89	14.69 **	3.26
*n*-6 PUFA	38.70	0.45	26.84 **	2.86	25.81 **	4.03
18:3*n*-3	0.47	0.12	0.31 **	0.03	0.26 **	0.03
20:5*n*-3	0.49	0.15	0.48	0.18	3.49 *** ^†^	1.03
22:5*n*-3	0.72	0.23	0.60	0.17	1.66 **^†^	0.38
22:6*n*-3	3.93	0.62	4.39	0.32	4.64	1.24
*n*-3 PUFA	5.89	0.69	5.87	0.69	9.18 *^†^	1.98
***n*-6 PUFA/*n-3* PUFA**	6.26	0.37	4.86 ***	0.02	2.74 ***^†^	0.43
**Total PUFA**	44.89	0.66	35.35 *	3.39	40.22 ^†^	3.29
**MUFA/SFA**	0.55	0.10	0.61	0.17	0.50	0.08
**Δ9-Desaturase index ^‡^**	0.58	0.07	0.62	0.09	0.54	0.05

CTRL, standard diet; HF, high-fat diet; HF-Phaeo, high-fat diet supplemented with 12% of *P. tricornutum*; MUFA, mono-unsaturated fatty acids; ND, not detected; PUFA, polyunsaturated fatty acids; SFA, saturated fatty acids. Results are represented as mean values ± SD, *n* = 6. Mean values were significantly different from those of the CTRL group: * *p* < 0.05; ** *p* < 0.001; *** *p* < 0.0001 (one-way ANOVA; LSD post hoc test). ^†^ Significant difference compared with the HF group (*p* < 0.05). ^‡^ Δ9-Desaturase index = [18:1(n-7 + n-9)/ 18:0 + 18:1(n-7 + n-9)].

**Table 3 nutrients-11-01069-t003:** Fatty acid composition of red blood cells phospholipids.

	CTRL		HF		HF-Phaeo	
Fatty Acids (mol %)	Mean	SD	Mean	SD	Mean	SD
**SFA**						
14:0	0.80	0.22	0.90	0.27	1.13	0.26
16:0	40.24	2.68	28.79 **	2.61	34.12 **^†^	5.04
18:0	14.86	2.64	13.90	1.58	15.03	1.69
Total SFA	55.91	3.90	43.41 **	2.02	50.27 ^†^	6.19
**MUFA**						
16:1	1.03	0.23	0.96	0.28	1.09	0.17
18:1 (*n*-7 + *n*-9)	11.64	1.30	14.01 *	2.03	11.94	0.59
20:1*n-9*	0.30	0.015	0.19 ***	0.04	0.12 ***^†^	0.02
22:1	0.76	0.06	0.97	0.25	0.24 ***^†^	0.09
24:1*n-9*	ND	-	ND	-	ND-	
Total MUFA	13.76	2.08	15.61	2.42	12.88	0.15
**PUFA**						
18:2*n*-6	6.69	0.55	7.69	1.04	6.05 ^†^	1.00
20:2*n*-6	0.37	0.02	0.84 **	0.17	0.48 ^†^	0.15
20:4*n*-6	10.48	1.83	17.05 **	3.45	8.81	3.09
*n*-6 PUFA	17.54	1.51	25.63 ***	3.50	13.74 ***^†^	1.92
18:3*n*-3	ND	-	ND	-	ND-	
20:5*n*-3	0.19	0.01	0.35	0.13	1.69 ***^†^	0.61
22:5*n*-3	3.28	0.87	2.83	0.46	11.39 ***^†^	2.21
22:6*n*-3	1.17	0.12	2.43 ***	0.41	1.59 ^†^	0.68
*n*-3 PUFA	10.38	1.91	5.79 ***	0.48	14.68 ***^†^	2.44
***n*-6 PUFA/*n-3* PUFA**	1.62	0.30	4.50 ***	0.62	1.12 ^†^	0.08
**Total PUFA**	26.85	2.21	31.81	4.44	30.02	5.84
**MUFA/SFA**	0.25	0.05	0.36 *	0.04	0.25 ^†^	0.00
**Δ9-Desaturase index ^‡^**	0.45	0.06	0.50	0.06	0.44	0.03

CTRL, standard diet; HF, high-fat diet; HF-Phaeo, high-fat diet supplemented with 12% of *P. tricornutum*; MUFA, mono-unsaturated fatty acids; ND, not detected; PUFA, polyunsaturated fatty acids; SFA, saturated fatty acids. Results are represented as mean values ± SD, *n* = 6. Mean values were significantly different from those of the CTRL group: * *p* < 0.05; ** *p* < 0.001; *** *p* < 0.0001 (one-way ANOVA; LSD post hoc test). † Significant difference compared with the HF group (*p* < 0.05). ‡ Δ9-Desaturase index = [18:1(n-7 + n-9)/ 18:0 + 18:1(n-7 + n-9)].

**Table 4 nutrients-11-01069-t004:** Fatty acid composition of liver phospholipids.

	CTRL		HF		HF-Phaeo	
Fatty Acid (mol %)	Mean	SD	Mean	SD	Mean	SD
**SFA**						
14:0	0.18	0.05	0.41 ***	0.05	0.32 ***^†^	0.04
16:0	20.94	1.08	18.91 *	1.48	18.45 *	1.16
18:0	17.41	2.21	19.58	1.01	21.95 *	2.47
Total SFA	38.53	1.83	38.90	1.74	40.72	1.55
**MUFA**						
16:1	1.09	0.39	0.84	0.19	0.78	0.23
18:1(*n*-7 + *n*-9)	7.74	1.43	9.42 *	0.68	7.29 ^†^	0.72
20:1*n-9*	0.22	0.06	0.14	0.07	0.12 *	0.06
22:1	0.26	0.05	0.39	0.10	0.20 ^†^	0.16
24:1*n-9*	ND	-	ND	-	ND	-
Total MUFA	9.27	1.88	10.72	0.99	8.38 ^†^	1.02
**PUFA**						
18:2*n*-6	11.15	1.08	7.76 **	1.49	7.89 **	1.12
20:2*n*-6	ND	-	ND	-	ND	-
20:4*n*-6	28.07	2.29	26.44	2.04	24.16 *	1.38
*n*-6 PUFA	39.22	2.06	34.21 ***	1.47	32.05 ***	1.65
18:3*n*-3	ND	-	ND	-	ND	-
20:5*n*-3	0.19	0.05	0.42	0.10	2.29 ***^†^	0.23
22:5*n*-3	0.90	0.11	0.98	0.17	2.43 ***^†^	0.28
22:6*n*-3	8.78	1.11	10.00	1.01	9.99	0.72
*n*-3 PUFA	9.88	1.25	11.31	0.82	14.72 ***^†^	1.03
***n*-6 PUFA/*n-3* PUFA**	4.04	0.72	3.04 ***	0.28	2.19 ***^†^	0.25
**Total PUFA**	49.10	1.17	45.52 **	1.46	46.76 **	1.09
**MUFA/SFA**	0.24	0.06	0.28	0.04	0.21 ^†^	0.03
**Δ9-Desaturase index ^‡^**	0.31	0.07	0.32	0.02	0.25 ^†^	0.04

CTRL, standard diet; HF, high-fat diet; HF-Phaeo, high-fat diet supplemented with 12% of *P. tricornutum*; MUFA, mono-unsaturated fatty acids; ND, not detected; PUFA, polyunsaturated fatty acids; SFA, saturated fatty acids. Results are represented as mean values ± SD, *n* = 6. Mean values were significantly different from those of the CTRL group: * *p* < 0.05; ** *p* < 0.001; *** *p* < 0.0001 (one-way ANOVA; LSD post hoc test). † Significant difference compared with the HF group (*p* < 0.05). ^‡^ Δ9-Desaturase index = [18:1(n-7 + n-9)/ 18:0 + 18:1(n-7 + n-9)].

**Table 5 nutrients-11-01069-t005:** Plasma biochemical parameters and liver lipid content.

	CTRL		HF		HF-Phaeo	
Parameter	Mean	SD	Mean	SD	Mean	SD
**Plasma biochemical parameters**						
ASAT (UI/L)	61.74	4.25	50.97 ***	5.71	52.75 ***	4.27
ALAT (UI/L)	41.52	5.16	50.30 ***	4.91	47.07 ***	6.49
ASAT/ALAT ratio	1.55	0.20	1.12 ***	0.11	1.09 ***	0.17
Glucose (mmol/L)	8.79	1.48	9.81 *	0.84	9.76 *	0.84
Insulin (μUI/mL)	37.68	8.26	89.17 ***	12.95	48.60 ***^†^	12.99
Leptin (ng/mL)	2.02	0.52	3.87 ***	0.53	2.25	0.53
AIP	0.44	0.09	0.72 ***	0.10	0.20 ***^†^	0.05
HOMA-IR	0.97	0.34	2.80 ***	0.57	1.09	0.22
**Liver lipids**						
TAG (mg/g)	65.99	11.74	153.58 ***	15.65	71.05 ^†^	13.34
TC (mg/g)	10.74	1.20	46.74 ***	2.99	11.70 ^†^	3.20

AIP, atherogenic index of plasma [log(TAG/HDL-C)]; ALAT, alanine amino-transferase; ASAT, aspartate amino-transferase; CTRL, standard diet; HF, high-fat diet; HF-Phaeo, high-fat diet supplemented with 12% of *P. tricornutum*; HOMA-IR, homeostasis model assessment of insulin resistance; TAG, triacylglycerol; TC, total cholesterol. Results are represented as mean values ± SD, *n* = 6. Mean values were significantly different from those of the CTRL group: * *p* < 0.05; *** *p* < 0.0001 (one-way ANOVA; LSD post hoc test). ^†^ Significant difference compared with the HF group (*p* < 0.05).

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
