# Peer review of "Preventive Effects of the Marine Microalga Phaeodactylum tricornutum, Used as a Food Supplement, on Risk Factors Associated with Metabolic Syndrome in Wistar Rats"

_nutrients, 2019, doi:10.3390/nu11051069_

Round 1
Reviewer 1 Report
The study highlighted the benefits of the marine microalga Phaeodactylum tricornutum as a diet supplement on improve the status of metabolic syndrome in a rodent model. The results show that a 8-week supplementation of the EPA-enriched marine microalgae with a high-fat diet favorably reduced metabolic syndrome related biomarkers such as plasma total cholesterol, triglycerides, TNF-alpha, and a few interleukins.
The study is well-written and the content is well-organized, yet a few spellings need to be fixed.
Please consider the following questions and provide feedback.
In Discussion (page 320-336), the authors mentioned that EPA is related to weight loss; however, the explanation of possible mechanism of action is missing. Please provide additional information there.
Safety - The study evaluated the changes of a few liver enzymes, but not yet reported relevant safety review. Overall, is this supplement safe to animals?
Author Response
Answers to reviewer 1 comments
Dear reviewer,
Please find below the answers to your comments :
Point 1: The study is well-written and the content is well-organized, yet a few spellings need to be fixed.
English language was checked carefully and several mistakes have been corrected.
Point 2: In Discussion (page 320-336), the authors mentioned that EPA is related to weight loss; however, the explanation of possible mechanism of action is missing. Please provide additional information there
The explanation of possible mechanisms has been added in the discussion section, page 15, lines 350 to 354.
Point 3: Safety - The study evaluated the changes of a few liver enzymes, but not yet reported relevant safety review. Overall, is this supplement safe to animals?
Since the goal of our study was to understand the impact of P. tricornutum supplementation in HF diet induced metabolic syndrome and not to develop a food supplement, toxicity / safety of the P. tricornutum supplementation was not extensively studied. According to the literature, ASAT and ALAT transaminases assays are commonly used as markers of hepatotoxicity and our results indicate the absence of important liver toxicity. We did not encountered any problems during the animal experiment. After the third day of food adaptation, the rats had well adapted to the diet without developing food allergies or intolerances with P. tricornutum. In addition, the rats fed with P. tricornutum supplementation do not have change in behaviour; they were sociable, reactive and not nervous. No alterations of food intake, unusual body growth, reduced activity, diarrhea, bleeding and death have been observed.
However, extensive toxicity studies in rats would be required in the context of the development of food supplements for human applications.

Reviewer 2 Report
Claire Mayer et al., in an animal model, studied the action of the marine microalga Phaeodactylum tricornutum in countering the development of the metabolic syndrome. The importance and innovation in studying the effects of micronutrients is linked to pollution of the seas and to the fish resources decrease, as clearly reported by the authors both in the abstract and in the introduction (58-77 lanes).
I congratulate Claire Mayer and her colleagues on good work: data are numerous and consistent with initial hypothesis and aims. In concise but careful introduction, the authors highlighted almost all aspects of the study. Materials and methods are extremely clear and detailed as well as results. A particular mention especially to the discussion: it is powerfully argued and it is indeed worthy of debate.
There is an only point that still need to be addressed: the use of HF diet implemented with 10% of fructose. This choice is very interesting, but in my opinion, the authors should respond better their choice. It is known that the abuse of this sugar becomes an extremely serious health problem (PMID: 28555043, PMID: 29408694, PMID: 29866605). Fructose plays a fundamental role in the development of insulin resistance and non-alcoholic hepatic steatosis (NAFLD). Above all, NAFLD progression in NASH is due to inflammatory mediators and oxidative stress (PMID: 30343320). Based on Phaeodactylum tricornutum properties and fructose role it is possible to speculate that this micronutrient can partly counteract the damage induced by fructose. The study of the action of this microalga and its possible use could therefore be even more interesting.
I suggest that the authors describe this aspect already in introduction and emphasize fructose use materials and methods to this choice. Furthermore, the authors could also calculate the HOMA index considering that they already have glucose and fasting insulin values after rats were fasted: this result could further enrich the work.
Finally, in the discussion, the authors can extend the period dedicated to fructose, in lanes 337-352 or 405-418 fructose and NAFLD. As suggested in other sections the use of this microalga could be more effective in association with other nutraceuticals, the authors could be speculate microalga action on the axis fructose-IR-NAFLD axis. For example, proposing the use of this nutraceutical in association with betaine or ferulic acid, that act not only in liver but also in muscle, other insulin tissue (PMID: 30047921, PMID: 23870626, PMID: 29260581, PMID: 28954428). This would highlight an important action of the compound as a molecule able to counteract the action of fructose. This aspect could then be investigated later in other works (please, the authors could be shorten nonalcoholic fatty liver disease with NAFLD and not with NFLD).
Author Response
Answers to Reviewer 2 comments
Dear reviewer,
Please find below the answers to your comments:
Point 1: There is an only point that still need to be addressed: the use of HF diet implemented with 10% of fructose. This choice is very interesting, but in my opinion, the authors should respond better their choice. It is known that the abuse of this sugar becomes an extremely serious health problem (PMID: 28555043, PMID: 29408694, PMID: 29866605). Fructose plays a fundamental role in the development of insulin resistance and non-alcoholic hepatic steatosis (NAFLD). Above all, NAFLD progression in NASH is due to inflammatory mediators and oxidative stress (PMID: 30343320). Based on Phaeodactylum tricornutum properties and fructose role it is possible to speculate that this micronutrient can partly counteract the damage induced by fructose. The study of the action of this microalga and its possible use could therefore be even more interesting. I suggest that the authors describe this aspect already in introduction and emphasize fructose use materials and methods to this choice.
According to the reviewer’s suggestion, NAFLD was introduced pages 2 and 3, lines 46 to 52 and 93 to 98; the use of Fructose was justified in the Material and methods section, page 3, lines 122 to 125; and both were discussed in the Discussion section, page 17, lines 444 to 459.
Point 2: Furthermore, the authors could also calculate the HOMA index considering that they already have glucose and fasting insulin values after rats were fasted: this result could further enrich the work
Thank you for this interesting suggestion. The HOMA-IR index was calculated as described in the Materials and methods section, page 4 lines 177 to 179; Results are presented page 11 lines 289 to 290 and in Table 5 p12; and finally HOMA-IR index results were noticed in the Discussion section, page 17 lines 459 to 461.
Point 3: Finally, in the discussion, the authors can extend the period dedicated to fructose, in lanes 337-352 or 405-418 fructose and NAFLD. As suggested in other sections the use of this microalga could be more effective in association with other nutraceuticals, the authors could be speculate microalga action on the axis fructose-IR-NAFLD axis. For example, proposing the use of this nutraceutical in association with betaine or ferulic acid, that act not only in liver but also in muscle, other insulin tissue (PMID: 30047921, PMID: 23870626, PMID: 29260581, PMID: 28954428). This would highlight an important action of the compound as a molecule able to counteract the action of fructose. This aspect could then be investigated later in other works
Thank you for this interesting suggestion. In perspective, the effects of microalga extracts associated with nutraceutical compounds (anthocyanins, phenolic acis) on NAFLD and IR have been developed in the conclusion section page 18 lines 504 to 511.

Reviewer 3 Report
This is a comprehensive study designed to evaluate the effects of the marine diatom P. tricornutum, as a source of long-chain omega-3 PUFA, on a constellation of metabolic factors in a preclinical model (high-fat and fructose fed Wistar rat) of metabolic syndrome. Overall the approach and study methods are sound and results clearly described. However, the discussion lacks focus and the interpretation tends to over-extend the results of this preclinical study. Specific comments are below:
Focus and simplify/shorted the Discussion: The results of the current study are not unexpected and are similar to what is described in the considerable literature on LC n-3 PUFA from fish oils (and to a lesser extent, other microalgae) in similar preclinical models of metabolic syndrome. Where this is true, this should be simply stated and referenced accordingly. Where this is not true, ie., where the results differ from what was expected, then this is where the discussion of the potential for additional contribution of other components of the microalgae of the e.g., carotenoids, pigments, fibers, etc., should be developed.
Refrain from comparing the results of this study to any clinical outcomes e.g., line 371; line 387; line 398; line 405…
Be consistent in descriptions of dose, for example, line 329 (36 g/kg of diet? Of body weight?); line 409 (101 mg/kg of diet? Of body weight? EPA+DHA).
There were differences observed in intake between the HF and HF-Phaeo at several timepoints across the 8 week study in which HF-Phaeo was lower – this observation should be discussed in light of possible impact on body weight/fat mass and metabolic outcomes.
Line 354: The relevance of this statement is unclear: The significant n-3 LC-PUFA level incorporation was observed in all organ tissues suggesting that the final conversion of docosapentaenoic acid (DPA) to DHA is the limiting factor in the conversion of EPA to DHA [38].
Line 429: is unclear: It means the EPA rich microalga O. aurita decreased the insulin plasma level and the level of IL-6 and interleukin-1 beta (IL-1β) pro-inflammatory cytokines in Wistar rats [61].
Line 466: the relevance of this statement is unclear: Moreover, our housing conditions did not introduce any bias and have allowed to obtained homogenous results.
Author Response
Answers to reviewer 3 comments
Dear reviewer,
Please find below the answers to your comments:
Point 1: Focus and simplify/shorted the Discussion: The results of the current study are not unexpected and are similar to what is described in the considerable literature on LC n-3 PUFA from fish oils (and to a lesser extent, other microalgae) in similar preclinical models of metabolic syndrome. Where this is true, this should be simply stated and referenced accordingly. Where this is not true, ie., where the results differ from what was expected, then this is where the discussion of the potential for additional contribution of other components of the microalgae of the e.g., carotenoids, pigments, fibers, etc., should be developed.
It is a fact that the effects of n-3 LC-PUFA from fish oils or microalgae have been extensively discussed in the literature in metabolic syndrome models. However, very few studies using microalgal biomass as food supplement have been published in the literature. The originality of our study is based both on the microalgae used, P. tricornutum, which has never been used in this context, and on the use of microalgal biomass as food supplement.
It is therefore necessary for each study mentioned to specify the strain, the form of use (oil, biomass, extracts), the dose and the duration of the treatment.
In order to shorten the discussion, some sentences, particularly those related to clinical studies, have been deleted.
Point 2: Refrain from comparing the results of this study to any clinical outcomes e.g., line 371; line 387; line 398; line 405…
These sentences comparing the results of this study to clinical outcomes have been deleted. Please, see in the discussion section page 18 lines 465 to 469, 482 to 484 and 501 to 503; p 19 lines 527-528.
Point 3: Be consistent in descriptions of dose, for example, line 329 (36 g/kg of diet? Of body weight?); line 409 (101 mg/kg of diet? Of body weight? EPA+DHA).
Dose descriptions have been clarified in the discussion section, pages 15 and 17 and lines 348 and 438.
Point 4: There were differences observed in intake between the HF and HF-Phaeo at several time points across the 8 week study in which HF-Phaeo was lower – this observation should be discussed in light of possible impact on body weight/fat mass and metabolic outcomes.
The expression of data on nutritional monitoring seems confusing. For greater clarity, the data have been changed and reported on body weight basis. The sentences have been modified in:
- Materials and methods section, page 3, lines 133 to 134
- Results section, page 5, lines 205 to 218
- Discussion section, page 15, lines 342 to 344
Indeed, this expression of data has brought more precision. The food intake varied with body weight as well as energy intake. It was observed (supplementary material, Figure S1) that HF-Phaeo rats had higher energy intake than other groups when data are reported on body weight basis. These observations were discussed in more detail to highlight the potential impact of bioactive molecules from microalga on energy expenditure, body weight and fat mass reduction (see Discussion section, page 15, lines 350 to 354 and 358 to 373).
Point 5: Line 354: The relevance of this statement is unclear: The significant n-3 LC-PUFA level incorporation was observed in all organ tissues suggesting that the final conversion of docosapentaenoic acid (DPA) to DHA is the limiting factor in the conversion of EPA to DHA [38].
The sentence has been modified to be more clear in the discussion section, page 16, lines 390 to 392.
Point 6: Line 429: is unclear: It means the EPA rich microalga O. aurita decreased the insulin plasma level and the level of IL-6 and interleukin-1 beta (IL-1β) pro-inflammatory cytokines in Wistar rats.
The sentence has been shortened (see discussion section, page 17, lines 463 to 465). In this previous study, it was observed a decrease of inflammation with O. aurita supplementation in Wistar rats fed with HF diet.
Point 7: Line 466: the relevance of this statement is unclear: Moreover, our housing conditions did not introduce any bias and have allowed to obtained homogenous results.
This sentence is in conformity with the NC3Rs ARRIVE Guidelines and underlines the fact that the variability of the data is not due to environmental factors as animal housing conditions. We do not think it is necessary to specify that in the text and the sentence has been deleted (see conclusion section, page 17, lines 572 to 575).
